

# Evaluation of ddRADseq for reduced representation metagenome sequencing

Michael Y. Liu,  Paul Worden,  Leigh G. Monahan,  Matthew Z. DeMaere,
Catherine M. Burke,  Steven P. Djordjevic,  Ian G. Charles  and  Aaron E. Darling

The ithree institute, University of Technology Sydney, Sydney, New South Wales, Australia

## ABSTRACT

**Background**.  Profiling of microbial communities via metagenomic shotgun sequencing has enabled researches to gain unprecedented insight into microbial community structure and the functional roles of community members. This study describes a method and basic analysis for a metagenomic adaptation of the double digest restriction site associated DNA sequencing (ddRADseq) protocol for reduced representation metagenome profiling.

**Methods**. This technique takes advantage of the sequence specificity of restriction endonucleases to construct an Illumina-compatible sequencing library containing DNA fragments that are between a pair of restriction sites located within close proximity. This results in a reduced sequencing library with coverage breadth that can be tuned by size selection. We assessed the performance of the metagenomic ddRADseq approach by applying the full method to human stool samples and generating sequence data.

**Results**. The ddRADseq data yields a similar estimate of community taxonomic profile as obtained from shotgun metagenome sequencing of the same human stool samples. No obvious bias with respect to genomic $G + C$ content and the estimated relative species abundance was detected.

**Discussion**. Although ddRADseq does introduce some bias in taxonomic representation, the bias is likely to be small relative to DNA extraction bias. ddRADseq appears feasible and could have value as a tool for metagenome-wide association studies.

# INTRODUCTION

'Who is doing what' is the ultimate open question in microbiome studies. Shotgun metagenomics is often applied to gain knowledge of functional roles for bacteria in microbial communities, where the data can be used to predict protein encoding genes and enzymatic pathways present in the community, sometimes leading to testable hypotheses for microbial function. Advances in DNA sequencing technology and computing have dramatically accelerated the development of sequence-based metagenomics, which has been proposed by many scientists as a means to characterize the function of microbes in microbiomes (*Handelsman, 2004*; *Simon & Daniel, 2011*). Nevertheless, studies seeking to link microbial community phenotype to genes in the metagenome, e.g. metagenome-wide association studies (MGWAS) can require large numbers of samples to be processed. As a

Corresponding author
Aaron E. Darling,
aaron.darling@uts.edu.au

result, MGWAS on genetically diverse samples such as mammalian faeces and soil remain difficult or intractable, due to the prohibitive cost of shotgun metagenome sequencing to adequate depth.

Here, we investigate the potential of reduced representation sequencing to be used for low-cost metagenome profiling. We describe a metagenomic adaptation of ddRADseq (*Peterson et al., 2012*), a method for genotyping by sequencing studies of large and complex individual genomes (e.g. plants). This approach takes advantage of the sequence specificity of restriction endonucleases, where the genomic DNA is first fragmented by restriction digestion to construct a set of sequences that are flanked by the targeted restriction sites. This results in a sequencing library with complexity that is reduced roughly in proportion to the density of the restriction enzyme cut-sites, with a complexity that is tunable via fragment size selection. The design of this approach also includes a dual-index combinatorial barcoding approach allowing samples to be multiplexed in a single sequencing run. We demonstrate the method on human stool samples and compare the recovered taxonomic profiles to those obtained by standard metagenome shotgun sequencing on the same samples.

## MATERIALS & METHODS

Ethical approval for this study was obtained from the University of Technology Sydney Human Research Ethics Committee (Approval number: UTS HREC REF NO. 2014000448).

Three healthy adult stool samples were collected and the DNA extracted using an UltraClean Microbial DNA isolation kit (MO BIO Laboratories). The ddRADseq metagenome libraries were generated following the original ddRADseq protocol (*Peterson et al., 2012*), which uses a double restriction digest followed by size selection to construct a library containing only a defined subset of the total genomic DNA, with fine-scale control of library complexity available via size selection. Combinations of commercially available restriction enzymes were computationally evaluated for cut site density and other properties in a set of reference genomes chosen to reflect a wide range of G + C content. The evaluation suggested a short list of optimal enzyme combinations, and the combination of NlaIII and HpyCH4IV (New England BioLabs) was selected for their properties of buffer compatibility, insensitivity to *dam* methylation, overhang incompatibility, and heat sensitivity. This combination of restriction enzymes was predicted to minimise bias in the representation of individual species within the metagenomic community. The protocol employed a dual index approach for sample barcoding where each adapter carries a sample barcode compatible with standard Illumina multiplexing index reads (Fig. 1). Finally, a variable length region containing randomly synthesised nucleotides is included immediately downstream of the read priming site, to improve cluster calling and offer the potential to identify PCR duplicates via unique molecular identifiers (*Kivioja et al., 2011*). The full list of adapters designed and used in this study is outlined in the Supplementary Information.

To generate the ddRADseq libraries, 50 ng of DNA from each sample was used in a restriction double digest following the reaction conditions recommended by the enzyme manufacturer. A total of 5U of both enzymes were used in the reaction and subsequently

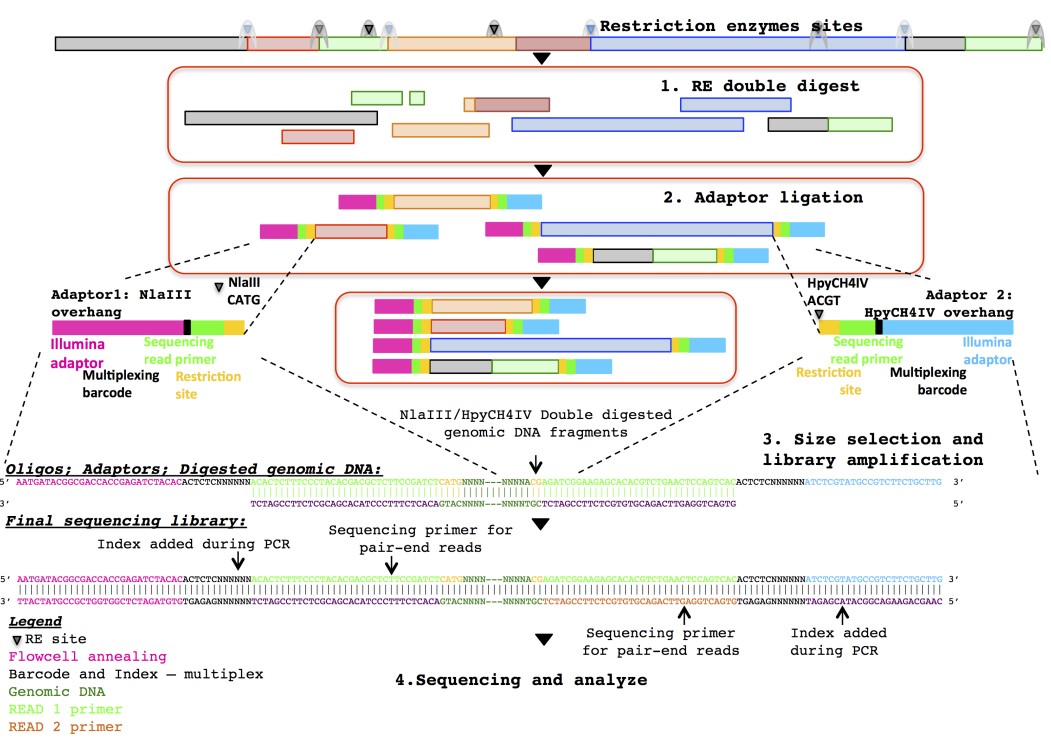

**Figure 1** Diagrammatic representation of the ddRADseq method: the oligos, adapters and final sequencing library.

heat inactivated following the manufacturer's instructions. A generous molar ratio of 1:40 (digested DNA: sequencing adapters) was then used for sequencing adapter ligation, which ensured an excess of adapters for ligation. Amplification of the adapter-ligated fragments was then carried out using the Illumina standard P5 and P7 flowcell oligo primers. The three sequencing libraries were then pooled with equal volume and size selected for approximately 500–600 bp fragments with SPRIselect beads (Beckman Coulter) using a 0.5×/0.6× double-sided clean-up. Shotgun metagenomic libraries were prepared from separate aliquots of sample DNA using the Illumina Nextera DNA kit. Sequencing of those samples were done with half a lane of the Illumina HiSeq 2500 in rapid PE250 mode. Details of the library construction protocol are given in the Supplementary Information.

## Quality assurance of sequencing data and availability of data

The quality of sequence data was examined using FastQC (*Andrews, 2010*). Reads were also checked for the presence of the sequenced portion of the restriction enzyme recognition site (NlaIII: CATG, HpyCH4IV: CGT) starting at position 0–3 in the read depending on the phasing of the barcode adapter associated with each sample. Shotgun sequence data are available from the NCBI Short Read Archive, project accession SRP100899, containing two datasets per sample.
## RESULTS

We present a full method for sample preparation using the double digest restriction site associated DNA sequencing (ddRADseq) technique. Paired-end Illumina sequencing was carried out on both shotgun and ddRADseq libraries for three human gut microbiome samples. As a pilot study with only three samples, there is insufficient statistical power to evaluate associations between phenotype and microbiome genotype in these samples. Instead, we evaluate the extent to which ddRADseq yields a similar estimate of community taxonomic profile as that obtained from shotgun metagenome sequencing of the same samples. If the ddRADseq is strongly biased against (or in favour of) some taxonomic groups it might reduce (or increase) power to detect associations between phenotype and the protein coding genes in the genomes of that group. Both the shotgun and ddRADseq reads were taxonomically profiled using MetaPhlAn (*Segata et al., 2012*), which counts reads matching to clade-specific protein-coding marker genes in order to estimate taxon abundance from metagenomic read data. The taxonomic profile between individuals was different on both the genus and species levels, consistent with the inter-individual variation that is typically observed in human microbiome studies. Highly similar, but not identical, taxonomic profiles were observed between the shotgun-sequenced and ddRADseq metagenome libraries. Hierarchical clustering using the Manhattan distance showed that ddRADseq samples cluster closely with their matching shotgun samples (Fig. 2). These results suggest that the representation of protein coding genes in ddRADseq metagenomes may be uniform enough to support association studies.

To assess whether the restriction enzyme pair used in this study created a reduced representation library that was biased by genomic G + C content we plotted the estimated relative abundance of each taxonomic group with respect to the genomic G + C content of a reference genome from that taxonomic group. As can be seen in Fig. 3, if an association exists between genomic G + C and relative abundance in our ddRADseq libraries, it is not visually apparent. We refrain from reporting Spearman correlations on this dataset, as such correlation tests have well known problems on compositional data (*Lovell et al., 2015*).

## DISCUSSION

Changes in the human microbiota have been associated with complex disease. The gut microbiota, for example, has been linked to conditions such as irritable bowel syndrome and inflammatory bowel disease. Major research efforts are now underway to move beyond taxonomic associations to discover and describe the genetic basis for these microbiome-associated diseases. Although shotgun metagenomics is one way to carry out those analyses, it can be prohibitively expensive in some situations. Therefore, reduced representation techniques such as ddRADseq may provide a useful means to enable large-scale studies within a fixed research budget. A metagenomic adaptation of the traditional ddRADseq protocol is feasible in principle, yet no data has previously been reported for such an application to real microbiome samples. Here we presented a preliminary analysis of the data and showed that metagenome profiling with ddRADseq appears to be feasible. Taxon relative abundances estimated from clade-specific marker genes are similar but

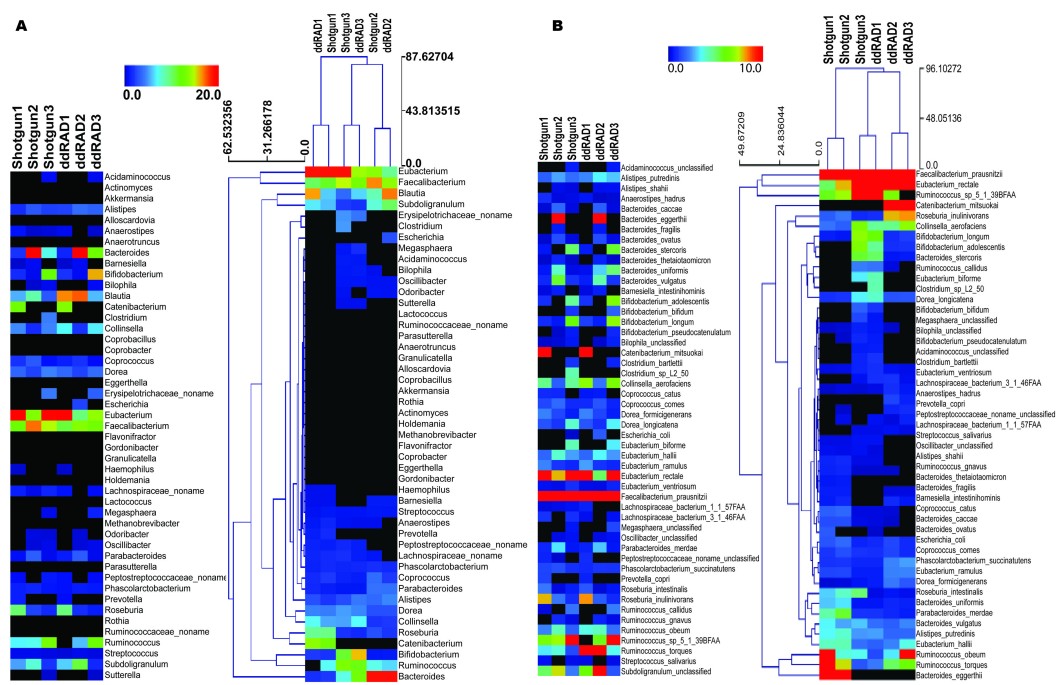

**Figure 2   Abundance of specific phylotypes detected in both shotgun and ddRADseq libraries.** (A) Samples clustered at the genus level; (B) Samples clustered at the species level. The heatmap is hierarchically clustered based on predicted abundances using the Manhattan distance metric.

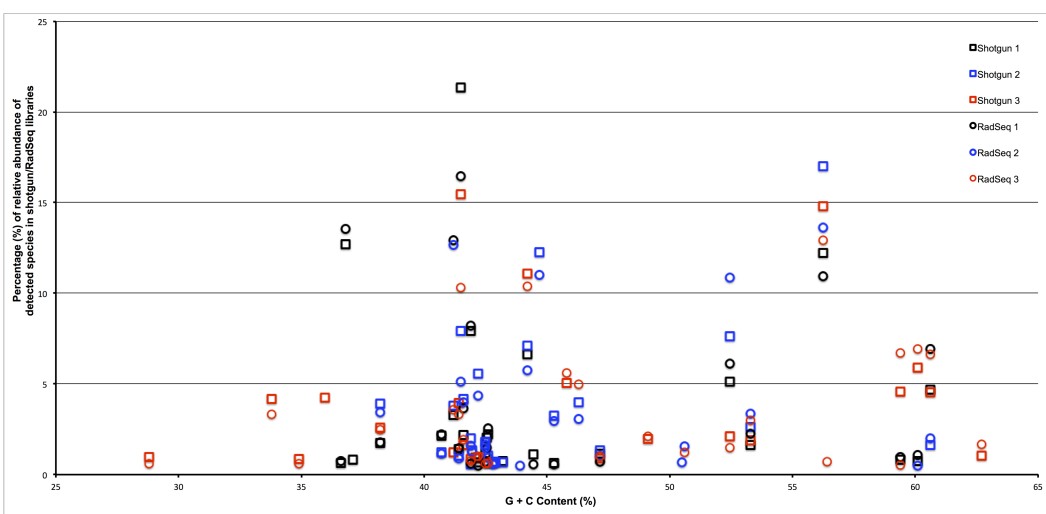

**Figure 3   Abundance of taxonomic groups in both shotgun and ddRADseq libraries as a function of genomic G + C content for all three samples.** Although differences in estimated relative abundance exist between shotgun and ddRADseq libraries, the ddRADseq protocol we used has not created a bias with an obvious association to genomic G + C content.

not identical to estimates from shotgun metagenomic data, implying that protein coding gene profiles from ddRADseq would likewise be similar to those obtained with shotgun metagenomics. The potential savings that could be obtained by the use of ddRADseq instead of shotgun metagenomics depends on several factors, including read length, community complexity, and the chosen degree of complexity reduction. To understand the potential cost savings, consider a set of high complexity samples (e.g. soil) to be sequenced on the highest output Illumina instruments, currently the NovaSeq 6000, which yield paired end 150 nt reads on fragments about 400 bp long. Assuming size-selection has been tuned to generate ddRADseq fragments about 300 nt long, with fragment start sites separated in the genomes by about 30,000 bp on average, this would represent a complexity reduction of 99%. A study which aims to associate markers with some phenotypic or other sample data would then in principle require 99% less data to find an association of equivalent statistical significance. In this example, 100 times as many samples could be processed with ddRADseq over shotgun metagenome sequencing, with equal statistical power and at equal cost in sequencing (though library preparation costs would increase). The trade-off is that phenotypic features are associated to markers, for example the abundance of individual ddRADseq fragments or single nucleotide variants within those fragments, rather than the potentially causal genotypic features themselves. Nevertheless, such markers may still be highly useful as they could provide a lead for further investigation of the features of genomes containing those markers. Therefore, when combined with careful fragment size selection, ddRADseq profiling could have value as a cost effective means to generate metagenomic profiles for use with metagenome-wide association studies and as a complementary tool for surveillance of microbial ecosystems, tracking differences across environments, treatments or time scale.

Even if enzyme choice leads to some bias in metagenomic ddRADseq libraries, other sources of bias may be more significant. For example, DNA extraction efficiency for individual community members can depend heavily on features such as the cell wall architecture of those organisms, leading to extreme bias in the extracted DNA and final sequencing library (*Frostegard et al., 1999*; *Morgan, Darling & Eisen, 2010*). The biases introduced by ddRADseq may be small in comparison to those introduced by DNA extraction. Sensitivity of restriction enzymes to methylation and other poorly understood DNA modifications maybe another factor contributing towards the observed differences in detection. The cleavage of target DNA may be blocked, or impaired, when a particular base in the enzyme's recognition site is modified. Future work to test this could explore the use of enzymes that have the same recognition site (isoschizomers), but with different methylation sensitivity, to identify modified bases through changes in the efficiency of restriction digest. Metagenomic ddRADseq will also be affected by biases that are commonly seen in single organism RADseq experiments including polymorphism in restriction site, mutation leading to novel restriction sites and heterogeneity in base composition across or between genomes (*Andrews et al., 2014*; *Andrews & Luikart, 2014*; *Arnold et al., 2013*; *DaCosta & Sorenson, 2014*; *Lowry et al., 2017*; *McCluskey & Postlethwait, 2015*; *Puritz et al., 2014*).

## CONCLUSIONS

This study presents a proof of concept for adopting double digest restriction site associated DNA sequencing (ddRADseq) against microbiome dataset. The generated sequencing results and analysis demonstrate: (1) the feasibility of adopting the ddRADseq profiling technique on complex microbiome samples; (2) the ability of this new technique to generate a reduced representation of a complex microbial community, and with taxon relative abundances that are similar to the community profile from shotgun metagenome sequencing of the same samples; (3) the reduced representation generated by the enzymes used in the ddRADseq method does not exhibit an obvious bias with respect to G + C content of genomes in the sample, which will support the wider application of such technique against other environmentally relevant samples. The method and data presented here may be of value to the research community, because the technique promises to enable metagenome-wide association studies in systems where it was previously intractable due to scale or complexity.

## ACKNOWLEDGEMENTS

We would like to acknowledge the Centre for Digestive Diseases in Sydney, NSW, Australia for their assistance with fecal sample collection and for helpful discussions.

### Funding

This research work was supported in part by the Australian Government through the Australian Research Council, Linkage grant LP150100912 and the AusGEM collaboration between UTS and the Department of Primary Industry of the New South Wales Government, Australia. There was no additional external funding received for this study. The funders had no role in study design, data collection and analysis, decision to publish, or preparation of the manuscript.

### Grant Disclosures

The following grant information was disclosed by the authors:
Australian Government through the Australian Research Council: LP150100912.
AusGEM collaboration between UTS and the Department of Primary Industry of the New South Wales Government, Australia.

### Competing Interests

The authors declare there are no competing interests.

### Author Contributions

- Michael Y. Liu performed the experiments, analyzed the data, wrote the paper, prepared figures and/or tables, reviewed drafts of the paper.
- Paul Worden, Leigh G. Monahan and Catherine M. Burke performed the experiments, reviewed drafts of the paper.

- Matthew Z. DeMaere analyzed the data, reviewed drafts of the paper.
- Steven P. Djordjevic contributed reagents/materials/analysis tools, reviewed drafts of the paper.
- Ian G. Charles conceived and designed the experiments, contributed reagents/materials/analysis tools, reviewed drafts of the paper.
- Aaron E. Darling conceived and designed the experiments, analyzed the data, contributed reagents/materials/analysis tools, wrote the paper, prepared figures and/or tables, reviewed drafts of the paper.

## Human Ethics

The following information was supplied relating to ethical approvals (i.e., approving body and any reference numbers):

Ethical approval for this study was obtained from the University of Technology Sydney Human Research Ethics Committee (Approval number: UTS HREC REF NO. 2014000448).

## DNA Deposition

The following information was supplied regarding the deposition of DNA sequences:

Shotgun sequence data are available from the NCBI Short Read Archive, project accession SRP100899, containing two datasets per sample.

## Supplemental Information

Supplemental information for this article can be found online at http://dx.doi.org/10.7717/peerj.3837#supplemental-information.

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
