# Peer review of "Evaluation of ddRADseq for reduced representation metagenome sequencing"

_PeerJ, doi:10.7717/peerj.3837_

## Round 0.1 · original submission · Minor Revisions

Please provide a point-by-point response to the reviewer's questions.
Also please bear in mind my own comments, listed here

1.The reported data represents (to our knowledge) the first metagenomic adaptation of the traditional ddRADseq protocol on real microbiome samples: Avoid claims of priority unless necessary to make your point.

2.When combined with careful fragment size selection, ddRADseq profiling could 190 have value as a cost effective means…: Please provide a tabulated breakdown of a cost comparison

3.Even if enzyme choice leads to some bias in metagenomic ddRADseq libraries, other sources of bias may be more significant: list the other sources WITH citations

4.intractable due scale or complexity: This is a sentence fragment
5.Fig 1 smallest fonts should be increased considerably

·

Basic reporting

No comment

Experimental design

No comment

Validity of the findings

The authors have used MetaPhlAn which uses clade-specific marker genes to classify sequencing reads so we know that protein coding genes are identified from the ddRADseq protocol, as also mentioned in the paper. The primary motivation for WGS metagenomics is the potential sampling of the entire genome instead of just a marker gene, for e.g. from a 16S survey. It would be useful if results from metagenomic assembly of reads from each sample using data from WGS and ddRADseq were compared. The reduced representation has been reported to skew results [1, 2] for certain analysis. Biases have been explored within the context of GC% but it will be worth evaluating the utility of sequences from ddRADseq protocol for metagenomic assembly

References:
1. Arnold, B., Corbett-Detig, R. B., Hartl, D., et al. (2013) Mol. Ecol., 22, 3179–3190, RADseq underestimates diversity and introduces genealogical biases due to nonrandom haplotype sampling.
2. Lowry, D. B., Hoban, S., Kelley, J. L., et al. (2017) Mol. Ecol. Resour., 17, 142–152, Breaking RAD: an evaluation of the utility of restriction site-associated DNA sequencing for genome scans of adaptation.

Additional comments

The authors propose a version of the double digest restriction site associated DNA sequencing (ddRADseq) protocol for metagenomic profiling. The primary advantage of this method over standard whole genome shotgun (WGS) metagenomics is the ability to profile a large number of samples at a relatively low cost. The restriction enzyme combination of NlaIII and HpyCH4IV is proposed after evaluating enzymes on a set of reference genomes with a range of GC%. The results from ddRADseq were shown to be comparable to those from whole genome shotgun analysis. The manuscript is well-written and presented clearly.

An approximate cost benefit analysis of ddRADseq and WGS for taxonomic identification will be useful for prospective users. The metrics for comparison can include the library preparation and sequencing costs besides qualitative information such as the number of species and abundance identified from each method.

Figure 3: It was a little difficult for me to interpret the plot with overlapping data points. It might help to use the same shape for shotgun and ddRADseq data points.

Reviewer 2 ·

Basic reporting

No comments

Experimental design

No comments

Validity of the findings

No comments

Additional comments

This study is a pilot work done on three individuals without any statistical significance. Although it is a preliminary analysis of the metagenomic profiling of the ddRADseq, I find this study relevant for large scale metagenome wide studies in a cost effective manner.
I have three suggestions:

1) Make the legend descriptive for Figure 2. There are two heat maps for both a and b. Please distinguish them.

2) There is no mention of supplemental information in the main article, please indicate in the methods section of the main manuscript that the authors have supplemental information as well.

3) The first two lines of the introduction are copy and paste of the abstract. Please don't use the same exact texts for both abstract and introduction, modify it accordingly.

---

## Round 0.2 · accepted · Accept

Thank you for your point-by-point responses to my comments and those of the reviewers.